# Mode of action of teixobactins in cellular membranes

Rhythm Shukla [1,2,8], João Medeiros-Silva [1,8], Anish Parmar[3,6,7], Bram J. A. Vermeulen [1], Sanjit Das[3,6,7], Alessandra Lucini Paioni[1], Shehrazade Jekhmane[1], Joseph Lorent [2], Alexandre M. J. J. Bonvin [1], Marc Baldus [1], Moreno Lelli[4], Edwin J. A. Veldhuizen[5], Eefjan Breukink [2], Ishwar Singh [3,6,7] & Markus Weingarth [1✉]

The natural antibiotic teixobactin kills pathogenic bacteria without detectable resistance. The difficult synthesis and unfavourable solubility of teixobactin require modifications, yet insufficient knowledge on its binding mode impedes the hunt for superior analogues. Thus far, teixobactins are assumed to kill bacteria by binding to cognate cell wall precursors (Lipid II and III). Here we present the binding mode of teixobactins in cellular membranes using solid-state NMR, microscopy, and affinity assays. We solve the structure of the complex formed by an improved teixobactin-analogue and Lipid II and reveal how teixobactins recognize a broad spectrum of targets. Unexpectedly, we find that teixobactins only weakly bind to Lipid II in cellular membranes, implying the direct interaction with cell wall precursors is not the sole killing mechanism. Our data suggest an additional mechanism affords the excellent activity of teixobactins, which can block the cell wall biosynthesis by capturing precursors in massive clusters on membranes.

[1] NMR Spectroscopy, Bijvoet Centre for Biomolecular Research, Department of Chemistry, Faculty of Science, Utrecht University, Padualaan 8, 3584 CH Utrecht, The Netherlands. [2] Membrane Biochemistry and Biophysics, Bijvoet Centre for Biomolecular Research, Department of Chemistry, Utrecht University, Padualaan 8, 3584 CH Utrecht, The Netherlands. [3] School of Pharmacy, JBL Building, University of Lincoln, Beevor St, Lincoln, UK. [4] Department of Chemistry 'Ugo Schiff', University of Florence, Via della Lastruccia 3, 50019 Sesto Fiorentino (FI), Italy. [5] Section Molecular Host Defence, Division Infectious Diseases & Immunology, Department Biomolecular Health Sciences, Faculty of Veterinary Medicine, Utrecht University, Yalelaan 1, 3584 CL Utrecht, The Netherlands. [6] Present address: Antimicrobial Pharmacodynamics and Therapeutics, Department of Molecular and Clinical Pharmacology, University of Liverpool, Sherrington Building, L69 3GA Liverpool, UK. [7] Present address: Department of Chemistry, The Robert Robinson Laboratories, University of Liverpool, L69 3BX Liverpool, UK. [8] These authors contributed equally: Rhythm Shukla, João Medeiros-Silva. ✉email: M.H.weingarth@uu.nl

Teixobactin, the first of a new class of antibiotics, kills a broad spectrum of clinically relevant multi-drug resistant pathogens, such as methicillin-resistant *Staphylococcus aureus* and *Mycobacterium tuberculosis* with excellent activity[1]. Teixobactin is a natural undecapeptide from *Eleftheria terrae* that comprises several uncommon residues including D-amino-acids and L-*allo*-enduracididine, with the last four residues forming a ring motif. Given that it inhibits bacterial cell wall synthesis by targeting conserved non-proteinogenic molecules (Lipid II and III) in the plasma-membrane (Fig. 1a)[1], development of resistance against teixobactin is difficult[2–4].

Notwithstanding the high therapeutic potential of teixobactin, progress towards its realization requires access to a large library of analogues due to high attrition rates in drug development[5] and shortcomings, such as difficult and prohibitively expensive

synthesis or insufficient solubility. Several research groups have addressed these shortcomings[6–10]. However, due to a fundamental lack of knowledge on the binding mode of teixobactin, the search for improved analogues has been largely based on coincidental discoveries. Structural insights on its binding mode are scarce[11,12] and reported data were collected in water or micelles using truncated Lipid II mimics, i.e., in non-physiological conditions. The lack of data in natural membrane settings is especially problematic because the binding mode and activity of Lipid II-binding antibiotics can be highly sensitive to the cellular environment[13].

Here we use solid-state NMR (ssNMR) spectroscopy, fluorescence microscopy, and affinity studies to determine at atomic resolution the native mode of action of the improved analogues[14] [R4L10]-teixobactin and [L10]-teixobactin (Fig. 1b and Supplementary Fig. 1)

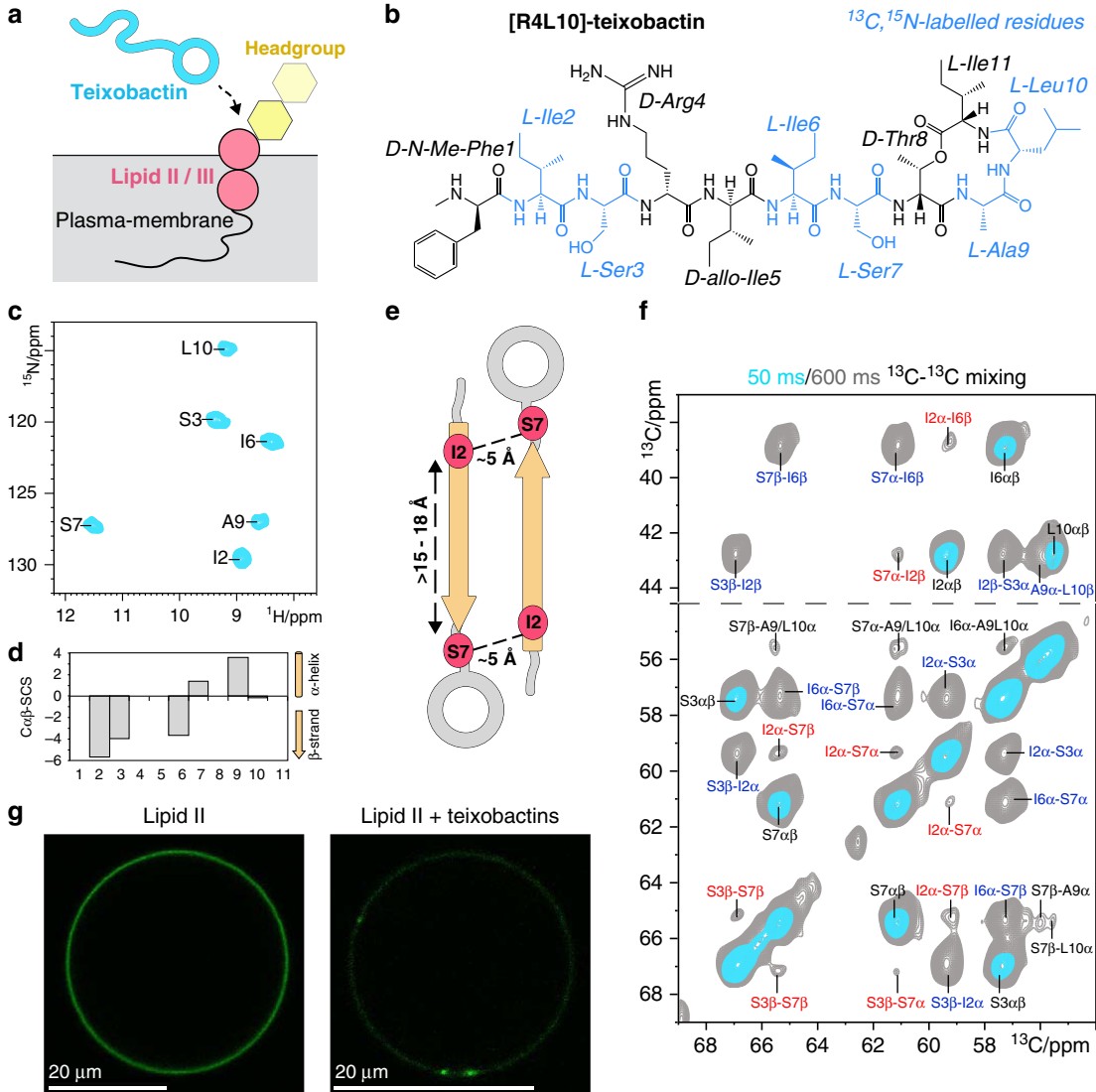

**Fig. 1 Teixobactins capture their target Lipid II in large clusters on the membrane. a** Teixobactins target cognate cell wall-precursors Lipid II and Lipid III in the bacterial plasma-membrane. **b** Chemical structure of [R4L10]-teixobactin in which D-Gln4 and L-*allo*-enduracididine10 of natural teixobactin are replaced by D-Arg4 and L-Leu10, respectively. $^{13}C,^{15}N$-labelled residues are coloured in blue. **c** 2D $^{15}N^1H$ ssNMR spectrum of Lipid II-bound [R4L10]-teixobactin showing all six $^{13}C,^{15}N$-labelled residues. **d** β-structuring of the N-terminus (Ile2–Ile6) of Lipid II-bound [R4L10]-teixobactin shown by Cαβ secondary chemical shifts (SCS)[15]. Source data are provided as a Source Data file. **e** Contacts observed in **f** between Ile2–Ser7 relate to intermolecular magnetization transfer between antiparallel β-strands. Distances are Cα–Cα spacings. **f** Overlay of 2D $^{13}C^{13}C$ PARISxy[48] spectra of Lipid II-bound [R4L10]-teixobactin measured with 50 (blue) and 600 ms (grey) magnetization transfer time. Intermolecular peptide–peptide contacts in red, sequential contacts in blue. All NMR spectra were acquired at 950 MHz. **g** Lipid II segregation caused by teixobactins visualized by fluorescence microscopy: GUVs doped with NBD-labelled Lipid II before (left) and after (right) treatment with [R4L10]-teixobactin.

in bacterial cellular membranes. We disclose that teixobactins trap Lipid II in μm-sized clusters on membrane surfaces as an alternative killing mechanism. We solve the structure of the complex between [R4L10]-teixobactin and Lipid II in these clusters, which provides detailed insights into the pharmacophore and explains how teixobactins mitigate antimicrobial resistance development. Unexpectedly, we find that the binding affinity of teixobactins for Lipid II is strongly attenuated if membranes are anionic, which is commonly the case for bacterial plasma-membranes. Together, our data raise astounding questions on the mode of action of teixobactins, which is more complex than previously thought.

## Results

**Teixobactins form large clusters on membrane surfaces**. We made [R4L10]-teixobactin amenable to ssNMR by synthetically[14] $^{13}$C,$^{15}$N-isotope labelling six residues (Ile2, Ser3, Ile6, Ser7, Ala9, Leu10) that cover the whole molecule. Co-assembly of [R4L10]-teixobactin and Lipid II in DOPC membranes resulted in well-resolved ssNMR correlation spectra that demonstrate the formation of a well-defined complex (Fig. 1c and Supplementary Fig. 2). We assigned the chemical shifts of [R4L10]-teixobactin using 2D $^{13}$C$^{13}$C and 3D CaNH and CONH experiments. The chemical shifts report on the secondary structure[15] and show that the N-terminus of Lipid II-bound [R4L10]-teixobactin adopts a β-strand conformation (Fig. 1d), which matches with its high rigidity (Supplementary Fig. 2). β-structuring was also observed in a teixobactin–sulfate complex in water[11] and in micelles with a truncated Lipid II variant[12]. This implies that β-structuring is a natural part of the binding mode that neither requires a membrane nor a complete receptor.

We hypothesized that β-structuring is caused by the oligomerization of Lipid II-bound teixobactins, in line with previously observed peptide aggregation behaviour in micelles[12]. We first examined the atomic-scale intermolecular arrangement of [R4L10]-teixobactins by a 2D $^{13}$C$^{13}$C ssNMR spectrum with a long magnetization transfer time that probes distances between $^{13}$C nuclei with a threshold of ~8 Å. We observed Cα–Cα contacts between residues Ile2–Ser7, Ile2–Ile6, and Ser3–Ser7 (Fig. 1e, f) that must stem from intermolecular magnetization transfer between antiparallel β-strands, given that the Cα$_i$–Cα$_{i+5}$ distance within the same β-strand is 15–18 Å. We confirmed the oligomerization with pyrene excimer fluorescence, which demonstrated that Lipid II molecules are within 1–2 nm of each other in the bound state (Supplementary Fig. 3)[16]. The oligomerization of Lipid II occurs virtually immediately after addition of [R4L10]-teixobactin, which suggests that antibiotic molecules need to come together to efficiently bind Lipid II, in agreement with the loss of activity of N-terminally truncated teixobactins that cannot oligomerise[17].

We used fluorescence microscopy and 7-nitrobenz-2-oxa-1,3-diazol-4-yl-(NBD)-tagged Lipid II in giant unilamellar vesicles (GUVs) to visualize the macroscopic state of bound teixobactins (Fig. 1g and Supplementary Fig. 4)[18]. To our surprise, we observed that Lipid II segregated into μm-sized clusters upon addition of [R4L10]-teixobactin, while it was initially homogeneously distributed over the GUV surface. We validated with [L10]-teixobactin, which is a close analogue of the natural antibiotic[1], that the segregation of Lipid II into clusters is a common property of teixobactins (Supplementary Fig. 4). While oligomerization is fast, cluster formation is a slower subordinate process that occurs on a timescale of about 2–4 h, which is the same timescale on which bacterial killing by teixobactins is observed[1,9,19]. Together, our data suggest that the capturing in large clusters makes Lipid II molecules unavailable for the

peptidoglycan biosynthesis, which is an alternative killing mechanism similar to lantibiotics[18].

**Complex interface and topology**. Teixobactins target Lipid II, a complex lipid with a pyrophosphate (PPi) and a headgroup composed of two sugars (MurNAc and GlcNAc) and a pentapeptide (Fig. 2c). We used ssNMR to determine the [R4L10]-teixobactin–Lipid II complex interface and topology in membranes.

We first examined how the antibiotic targets PPi. $^{31}$P ssNMR data in membranes show stark changes of the Lipid II PPi signals upon binding of [R4L10]-teixobactin, suggesting a direct interaction (Fig. 2a). Indeed, 2D $^{1}$H$^{31}$P ssNMR data established that backbone amino protons of [R4L10]-teixobactin directly coordinate the pyrophosphate (Fig. 2b). The interfacial correlations match with the amino protons of Ala9 and Leu10, showing that the ring motif of [R4L10]-teixobactin coordinates PPi. Conversely, we observed no correlation for Ser7, which means that Ser7 and the β-structured N-terminus are not part of the PPi-binding interface.

We next show how the Lipid II sugar-pentapeptide headgroup participates in the complex. For this, we developed an approach to $^{13}$C,$^{15}$N-label Lipid II, which has high potential to characterize the binding modes of other promising antibiotics[18,20,21]. We used 2D $^{13}$C$^{13}$C spectra to pinpoint the interface between $^{13}$C,$^{15}$N-[R4L10]-teixobactin and $^{13}$C,$^{15}$N-Lipid II (Fig. 2c, d and Supplementary Fig. 5). While we observed intense signals for the Lipid II sugars, those of the pentapeptide were faint or absent. Since we used dipolar-based ssNMR experiments in which only rigid residues give signals, our data imply that the pentapeptide is mobile and hence likely not part of the interface. This was confirmed with a scalar-based ssNMR experiment that enables detection of mobile residues, showing the missing pentapeptide signals (Supplementary Fig. 6). For the sugars, we obtained many interfacial contacts that all related to Ala9 and Leu10, confirming the interaction of the ring motif of [R4L10]-teixobactin with Lipid II (Fig. 2c). MurNAc, which is covalently attached to Lipid II PPi, shows a large number of contacts, implying its direct presence at the interface (Fig. 2e), whereas we obtained only one low-intensity unambiguous correlation with GlcNAc, suggesting that the second sugar is only remotely involved in the interface.

We determined the membrane-topology of Lipid II-bound [R4L10]-teixobactin using a mobility-edited ssNMR experiment[22] in which magnetization from mobile water or lipids is transferred to the rigid antibiotic (Supplementary Fig. 7). We found that [R4L10]-teixobactin localizes at the water–membrane interface with the non-polar residues Ile2 and Ile6 partitioned in the bilayer. Together, our data provide a detailed description of the complex in membranes (Fig. 2f).

**High-resolution structure of the complex**. We next calculated using HADDOCK[23] a complex of four [R4L10]-teixobactin and Lipid II molecules as minimal binding motif. The structure calculations were based on interfacial distance restraints between [R4L10]-teixobactin and Lipid II, distance restraints between [R4L10]-teixobactin monomers, and dihedral angle restraints. We note that our structural insights are limited by the lack of isotope-labels for 5 out of 11 residues in [R4L10]-teixobactin.

The obtained structures superimposed well (1.77 ± 0.49 Å RMSD for [R4L10]-teixobactin) (Fig. 3a and Supplementary Fig. 8) and reveal antiparallel [R4L10]-teixobactin fibre-like β-sheets. Pairs of β-strands are shifted with respect to each other, which leaves the ring motif free to bind Lipid II. The complex shows that the critical sequence of D- and L-amino acids in teixobactins[10] enables separation of hydrophilic and hydrophobic residues (Fig. 3c), i.e., sidechains of sequential pairs of D- and

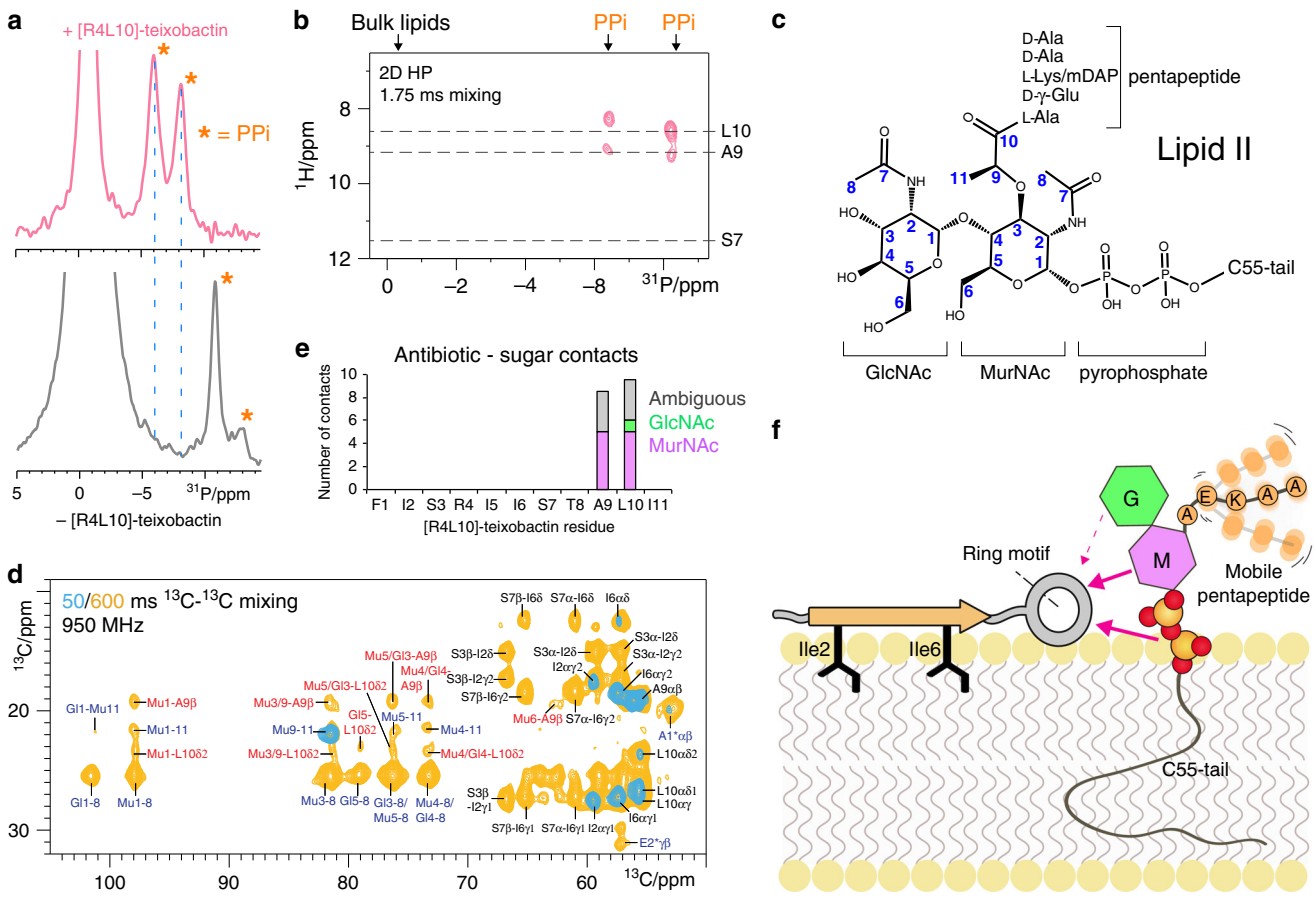

**Fig. 2 Binding to the pyrophosphate and first sugar of Lipid II is dominant. a** 1D $^{31}$P spectra of Lipid II in liposomes in the presence (pink) and absence (gray) of [R4L10]-teixobactin. Orange asterisks mark PPi signals; bulk lipids come around 0 ppm. **b** 2D $^{1}$H$^{31}$P spectrum of the complex. Dashed lines indicate chemical shifts of the amino protons of Ser7, Ala9, and Leu10. **c** Chemical model of Lipid II. **d** 2D $^{13}$C$^{13}$C PARISxy[48] spectra of the $^{13}$C,$^{15}$N-labelled complex measured using 50 (cyan) and 600 ms (yellow) magnetization transfer time. Contacts within and between [R4L10]-teixobactin molecules labelled in black, intramolecular Lipid II contacts in blue, interfacial contacts in red. See Supplementary Fig. 5 for the full spectrum. **e** Sum of interfacial contacts between [R4L10]-teixobactin and the sugars MurNAc (magenta) and GlcNAc (green); ambiguous contacts in gray. Source data are provided as a Source Data file. **f** Illustration of interface and topology of the complex. [R4L10]-teixobactin resides at the water–lipid interface with Ile2 and Ile6 partitioning into the membrane. The ring motif coordinates PPi. MurNAc (M, magenta) and, to a minor extent, GlcNAc (G, green) also interact with the ring motif. The mobile pentapeptide (orange) is not involved in the interaction.

L-residues (e.g., D-N-Me-Phe1 and L-Ile2/D-allo-Ile5 and L-Ile6) are located on the same side of the β-sheet. The long hydrophobic sidechains of N-Me-Phe1, Ile2, Ile5, and Ile6 align and act as membrane-anchors (Fig. 3d), in agreement with computational data[24], explaining why substitutions by polar[8,25] or short[26] sidechains or inversion from D to L configuration[10] lead to complete loss of activity. The only long hydrophobic sidechain that did not point towards the membrane-side is Ile11 (Supplementary Fig. 9), whose substitution by alanine is tolerated[26]. The hydrophilic Ser3, Arg4, and Ser7 are water-exposed and form hydrogen bonds that, enabled by the long Arg4 sidechain, connect three [R4L10]-teixobactin molecules (Supplementary Fig. 9).

The ring amino protons (Thr8-Ile11) and the cationic N-terminus of a neighbouring teixobactin molecule coordinate the pyrophosphate of Lipid II. This explains why the presence of a positively charged N-terminus is critical for the activity[27] and shows that dimer formation of teixobactins is required for efficient Lipid II binding (Fig. 3b). The MurNAc sugar also directly contacts the ring motif, while GlcNAc is distal from the drug and the mobile pentapeptide adopts random orientations. Non-polar groups of MurNAc face the hydrophobic

Leu10 sidechain, which possibly contributes to the good activity of teixobactins with branched-chain residues (Leu, Ile, Val) at position 10[14]. Diverse hydrophobic and cationic residues are however tolerated at positions 9 and 10 of the ring motif[10,17,26], which shows that interactions with MurNAc and GlcNAc are of low specificity. Analogously, our ssNMR data, in line with MD simulations[24], exclude specific interactions with the pentapeptide.

We validated that teixobactins are able to bind a broad spectrum of cognate cell wall precursors[1] by comparing (i) Lys-Lipid II, (ii) mDAP-Lipid II, and (iii) Lys-Lipid I. While Lipid I lacks the GlcNAc sugar, mDAP-Lipid II has a different pentapeptide. While Gram-positive bacteria have a cationic Lysine (Lys-Lipid II) in the pentapeptide, it is replaced by a zwitterionic *meso*-diaminopimelic acid (mDAP) in Gram-negative (mDAP-Lipid II) (Fig. 2c)[2,3]. As expected, [R4L10]-teixobactin bound to Lys-Lipid II or mDAP-Lipid II gave the same ssNMR spectra, in line with high mobility of the pentapeptide and the absence of interfacial contacts with the pentapeptide. [R4L10]-teixobactin was also able to bind Lys-Lipid I and exhibited clear signal changes in the ring motif (Fig. 4a and Supplementary Fig. 10), showing that GlcNAc modulates, presumably allosterically, drug binding.

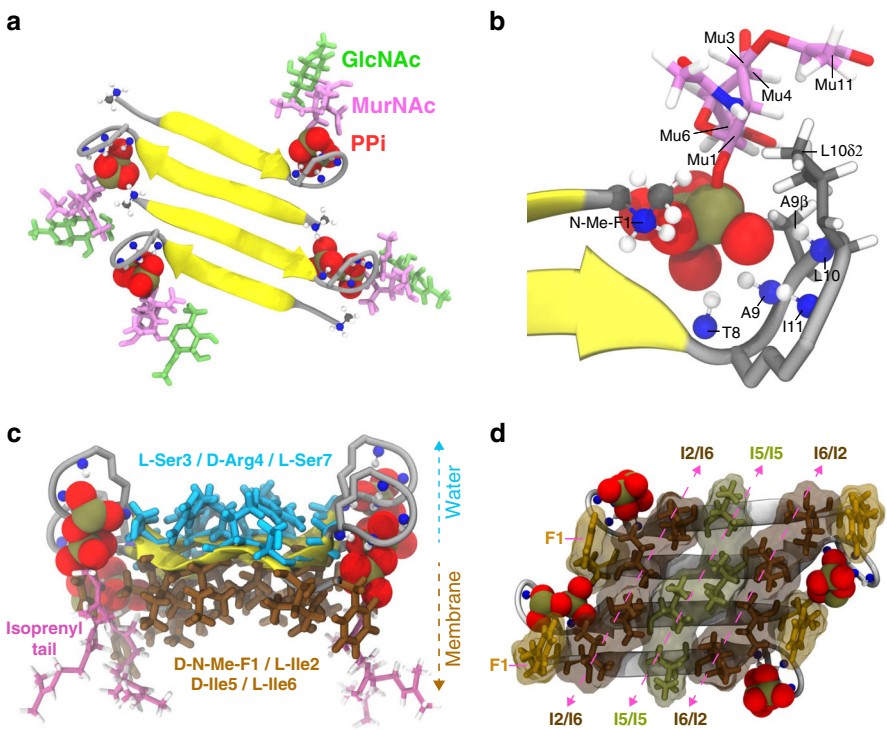

**Fig. 3 Structure of the [R4L10]-teixobactin–Lipid II complex in membranes. a** [R4L10]-teixobactin aligns into antiparallel fiber-like β-sheets (yellow). Lipid II PPi is shown in red, MurNAc in magenta, GlcNAc in green. Backbone nitrogens and amino protons that coordinate PPi are shown explicitly. **b** Lipid II binding site of [R4L10]-teixobactin. Amino protons of the ring motif and the N-terminus coordinate PPi, while nonpolar groups of MurNAc (in pink) face the Leu10 sidechain. **c** Hydrophilic residues (Ser3, Arg4, Ser7; in cyan) of [R4L10]-teixobactin are water-exposed; hydrophobic residues (N-Me-F1, Ile2, Ile5, Ile6; in brown) and the Lipid II isoprenyl tails point towards the membrane. **d** View from the membrane-site of [R4L10]-teixobactin: the membrane-embedded isoleucine residues align.

In summary, a picture of a balanced binding mode emerges in which teixobactins use the conserved pyrophosphate as firm, specific anchor, whereas the interaction with the sugar-pentapeptide headgroup is more loosely defined[28,29]. This behaviour enables teixobactins to recognize different cognate cell-wall building blocks and wall teichoic acids precursors, such as Lipid I, Lipid II, Lipid III, and undecaprenyl pyrophosphate that all have the PPi moiety in common but feature different headgroups[1]. The capacity to recognize several conserved precursor molecules both maximizes the number of targets and decreases the likelihood of the development of efficient resistance mechanisms against teixobactins. Taken together, a molecular rationale emerges for the broad activity spectrum of teixobactins and for their ability to bind, unlike vancomycin, Lipid II versions with different pentapeptides[30].

**Cellular conditions change the binding mode.** Eventually, we examined the impact of the membrane composition that can be critical for the native binding mode and is unknown for teixobactins. We probed the impact of the membrane charge with comparative ssNMR measurements in neutral DOPC and anionic DOPG liposomes (Fig. 4a, b). In anionic membranes, we observed widespread signal changes for residues Ile2, Ser7, Ala9, and Leu10, showing impact on both the N- and C-termini which teixobactins use to coordinate Lipid II. Unexpectedly, ssNMR signal intensity dropped by a factor of six in anionic membranes (Supplementary Fig. 11). Since ssNMR sensitivity may inversely correlate with molecular mobility, we hypothesized that the dynamics of the complex was augmented in anionic membranes. Indeed, ssNMR relaxation data confirmed a stark increase in mobility of Lipid II-bound [R4L10]-teixobactin with increasing anionic membrane

charge (Fig. 4c). This mobility increase is of global nature and similar for the entire peptide and the well-resolved residue Ser7. We validated with a second analogue, [L10]-teixobactin, that reduced ssNMR sensitivity and enhanced dynamics are common in anionic membranes (Supplementary Fig. 11).

The drop in sensitivity and the increase in dynamics likely point to a decreased binding affinity to Lipid II in anionic membranes. We compared binding affinities between [R4L10]-teixobactin and Lipid II in neutral and anionic membranes with isothermal titration calorimetry (ITC)[31]. We found a much weaker binding affinity in membranes containing anionic lipids, with an equilibrium dissociation constant $K_d$ that increased from 0.17 μM in neutral membranes to 21.9 μM in membranes containing 25% (mol/mol) of anionic lipids (Fig. 4d, e).

We hypothesized that the stark impact of the membrane composition should modulate the cellular binding mode of teixobactins, which is eventually decisive for drug design. SSNMR[32–35] enables high-resolution drug–receptor studies in crowded cellular conditions[13], but requires highly sensitive methods such as [1]H-detection[13,36] and dynamic nuclear polarization (DNP)[33,35,37,38] due to the minute native Lipid II concentration[39], which is maximally 0.5–1 % of the membrane lipids[40]. Here, we investigated the [R4L10]-teixobactin–Lipid II complex directly in natural membranes isolated from the Gram-positive bacterium *M. flavus*. We first show that the oligomerization of teixobactins and the complex structure are conserved in natural membranes, which is visible from the good match of DNP-ssNMR 2D [13]C[13]C spectra in cellular and in vitro (DOPC) conditions (Supplementary Fig. 12). Next, we sought to reveal an impact of the cellular environment with a well-resolved 2D NH spectrum (Fig. 4a, right panel). Intriguingly, the cellular 2D NH

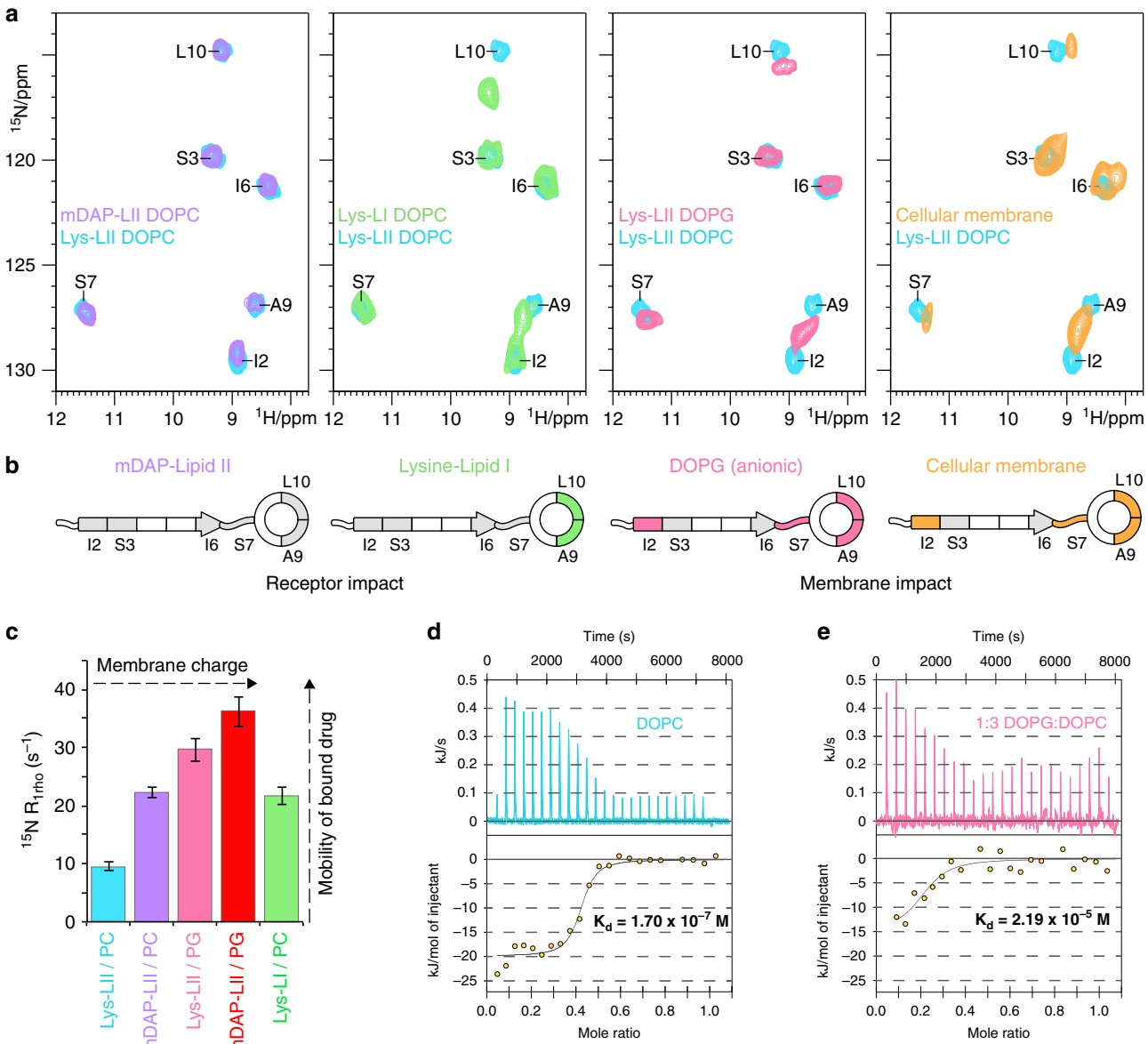

**Fig. 4 Teixobactins have low affinity for Lipid II in anionic membranes. a** 2D $^{15}$N$^{1}$H ssNMR spectra of [R4L10]-teixobactin bound to Lys-Lipid II in DOPC (cyan), mDAP-Lipid II in DOPC (violet), Lys-Lipid I in DOPC (green), Lys-Lipid II in DOPG (pink), and in cellular *Micrococcus flavus* membranes (orange). The cellular spectrum was measured with 7.5 nmol (<10 μg) of drug. **b** Illustration of spectral differences shown in **a**. HN CSPs >0.15 ppm are shown in colours, isotope-labelled residues with smaller CSPs in gray. **c** ssNMR microsecond dynamics of bound [R4L10]-teixobactin for different membrane and receptor types. The error bars show the standard error of the fit. Source data are provided as a Source Data file. **d** Representative isothermal titration calorimetry (ITC) data of [R4L10]-teixobactin and Lys-Lipid II in neutral DOPC and **e** anionic membranes, composed of a 1:3 (mol/mol) DOPG:DOPC mixture. ITC measurements were performed in triplicate for each system.

spectrum of bound [R4L10]-teixobactin strongly resembled the one in anionic membranes, in line with a high content of anionic lipids (up to 90%) in *Micrococcal* membranes[41]. Since the binding affinity to Lipid II is much lower in anionic membranes, our data imply that teixobactins bind Lipid II with considerably reduced affinity under cellular conditions given that bacterial plasma-membranes are negatively charged[41]. A molecular rationale for the observed effect could be the interaction of anionic lipids with the cationic N-terminal charge and the ring motif.

## Discussion

In this study, we have presented an in-depth analysis of the binding mode of teixobactins in cellular membranes. We have presented a high-resolution structure of a teixobactin-analogue in complex with Lipid II in lipid membranes (Fig. 3). The structure provides a wealth of information on the pharmacophore and demonstrates that teixobactins balance specific and fuzzy binding to recognize a maximal number of targets and to minimize the likelihood of resistance development. Furthermore, our structure explains the critical sequence of D- and L-amino acids that enables a clear-cut separation of hydrophilic and hydrophobic sidechains and thereby also provides a molecular rationale why certain positions do not tolerate substitutions of hydrophobic residues by hydrophilic residues and vice versa[8,10,25,26,42,43].

Besides the structure, our study presents two other major findings, which are that teixobactins form μm-sized clusters on membrane surfaces and that their binding affinity to Lipid II is

markedly reduced in anionic membranes. The impact of membrane lipids on teixobactins was previously not considered. The modest affinity for Lipid II in membranes that contain high amounts of anionic lipids is particularly surprising because most bacterial membranes are strongly anionic and teixobactins show high activity against a broad bacterial spectrum[1,14]. The direct binding of cell wall precursors is hence unlikely solely responsible for bacterial killing by teixobactins. Our data suggest that the subsequent slower formation of µm-sized clusters, that separates precursors from the cell wall biosynthesis and occurs on the timescale on which teixobactins kill bacteria, is an additional mode of action, and potentially even the more relevant one in physiological conditions. The relevance of cluster formation for activity is also congruent with studies of fluorescent teixobactins that showed pronounced staining of cell wall synthesis sites in Gram-positive bacteria[44]. Cell wall precursors presumably get effectively entangled in the massive clusters despite a relatively weak binding affinity to teixobactins in anionic cellular conditions. Indeed, we validated with fluorescence microscopy that teixobactins are able to efficiently capture Lipid II in clusters in anionic membranes (Supplementary Fig. 13). While cluster formation was previously only observed for lantibiotics[18], it is hence a more common mode of action of antibiotics that target cell wall precursors.

Future studies on how membrane lipids and cluster formation modulate the activity of teixobactins are urgently required to understand the mode of action of teixobactins. Our data also provide compelling evidence that such studies require physiologically relevant media and, ideally, cellular conditions[13]. In this regard, concerted optimization of cluster formation and of the binding affinity to different cell wall building blocks in relevant membrane conditions may lead to further improvements in the design of teixobactins. These insights may critically advance our capacity to develop potent antibiotics against multi-drug resistant bacteria.

## Methods

**Materials**. Phospholipids 1,2-dioleoyl-sn-glycero-3-phosphocholine (C18:1, DOPC) and 1,2-dioleoyl-sn-glycero-3-phosphoglycerol (C18:1, DOPG) were purchased from Avanti Polar Lipids, Inc.

**Isothermal titration calorimetry**. Preparation of liposomes: For ITC measurements large unilamellar vesicles (LUVs) containing Lys-Lipid II were prepared by incorporating 2% (mol/mol) of Lys-Lipid II in DOPC from the stock solution. The lipids were dried by nitrogen stream and hydrated with 10 mM HEPES, 150 mM NaCl, pH 7.5 buffer to a lipid-phosphate concentration of ~20 mM. LUVs were obtained after 10 rounds of extrusion through 200 nm membrane filters (Whatman Nuclepore, Track-Etch Membranes). The liposomes were dialysed to citrate phosphate buffer (100 mM NaCl, 50 mM citric acid, 100 mM $Na_2HPO_4$, pH ~6). Anionic LUVs were prepared by mixing 25% (mol/mol) DOPG and 75% (mol/mol) DOPC and then incorporating 2% (mol/mol) Lys-Lipid II. All other steps were the same as for zwitterionic LUVs.

ITC was performed with the Low Volume NanoITC (Waters LLC, New Castle, DE, USA) to determine interaction between LUVs and [R4L10]-teixobactin. [R4L10]-teixobactin was diluted in a citrate phosphate buffer (100 mM NaCl, 50 mM citric acid, 100 mM $Na_2HPO_4$, pH ~6) to a final concentration of 65 µM. Samples were degassed before use. The chamber was filled with 164 µL of [R4L10]-teixobactin, and the LUVs were titrated into the chamber at a rate of 1.96 µL/300 s with a stirring rate of 300 rpm. Experiments were performed at 37 °C and analysed using the Nano Analyse Software (Waters LLC). All experiments were performed in triplicate for each system. Control experiments were performed with Lipid II-free LUVs.

**Fluorescence microscopy**. GUVs were prepared using the electroformation method. The GUV cell is a self-assembled system which consists of a 5 mm-thick Teflon plate which is aligned with two titanium electrodes in a closed Teflon chamber. 1.5 µL of 0.5 mM DOPC doped with NBD-labelled Lipid II (0.1 mol%) were brushed on the electrodes. Next, the chamber was filled with 350 µL 0.1 M sucrose solution; the electrodes dipped in and then connected to a power supply which generated a sine wave of voltage 2.5 V at a frequency of 10 Hz for 90 min.

Each microscopy slide (µ-slide 8 well, Ibidi) was covered with 350 µL BSA solution (1 mg/mL). To detach the GUVs from the electrodes the power supply was changed to square wave of voltage 2 V at a frequency of 2 Hz for 15 min. The slides were immersed in 300 µL 0.1 mM glucose solution to which 80 µL of GUVs were added. These were incubated for 4 h with 1 µM [R4L10]-teixobactin and observed by confocal microscopy. Anionic GUVs were prepared with a homemade electroformation setup with ITO coverslips (18 × 18 mm, 15–30 Ω, SPI supplies). On both slips, 3 µL of a chloroform–methanol (1:1, vol/vol) solution containing 1 mM of a 3:1 mixture of DOPC:DOPG (mol/mol) and 0.1 mol% NBD-labelled Lipid II was deposited and dried in a vacuum desiccator. With these slips, the electroformation chamber was assembled and filled with 0.1 M sucrose solution. The GUV cell was connected to the power supply and an alternating current applied (1 V, 10 Hz). GUV formation was carried out overnight. The GUVs were then incubated with 1 µM [L10]-teixobactin for 4 h in a 0.1 M glucose solution.

Microscopy: GUVs were imaged using Zeiss LSM 880 with ×63/1.2NA glycerol objective lens. The NBD label appeared green upon excitation by the 488 nm laser. The emission range for detection was 530–545 nm ($\lambda_{em\ peak}$ = 539 nm). The brightfield images were used for detection and location of the GUVs without a fluorescent signal and also as a control to observe their shape. FIJI software was used for the image analysis[45].

**Fluorescence spectroscopy**. The oligomerisation of fluorescent pyrene-labelled Lipid II was observed upon binding of [R4L10]-teixobactin. DOPC LUVs containing 0.5 (mol%) of pyrene-labelled Lipid II in buffer (10 mM Tris–Cl, 100 mM NaCl, pH 8.0) were prepared as described above. All fluorescence experiments were performed with a Cary Eclipse (FL0904M005) fluorometer[46]. All samples (1.0 mL) were continuously stirred in a 10 × 4-mm quartz cuvette and kept at 20 °C. [R4L10]-teixobactin was titrated to the LUVs. Pyrene fluorescence was followed with spectral recordings between 360 and 550 nm ($\lambda_{ex}$ 350 nm, bandwidth 5 nm). The emission at 380 and 495 nm was recorded and averaged over 50 s, to obtain the values for the monomer and excimer intensity, respectively, to determine the excimer over monomer ratio for all conditions.

**Solid-state NMR spectroscopy**. 1D, 2D, and 3D $^1H$-detected experiments were performed at 60 kHz MAS frequency using magnetic fields of 16.4, 18.8, and 22.2 T (700, 800, and 950 MHz $^1H$-frequency) at temperatures of 305 K. All $^1H$-detected experiments were dipolar-based. PISSARRO low-power decoupling[47] was used during detection periods. 2D $^{13}C^{13}C$ spin diffusion experiments were performed with PARISxy[48] recoupling at 950 MHz and 15.5, 17, or 18 kHz MAS frequency at 270 K temperature using mixing times of 25–1000 ms. SPINAL64[49] was used in detection periods. The 2D scalar $^{13}C^{13}C$ TOBSY[50] experiment was acquired at 950 MHz with 8 kHz MAS at 295 K temperature with 6 ms $^{13}C$–$^{13}C$ magnetization transfer time. Chemical shift assignments were performed using standard $^1H$-detected 3D CANH and CONH experiments[36,51] in combination with 2D $^{13}C^{13}C$ PARISxy[48] experiments with different magnetization transfer times (see Supplementary Tables 1 and 2). DNP-enhanced 2D $^{13}C^{13}C$ spin diffusion (20 ms mixing time) ssNMR experiments[52] were carried out using a 263 GHz microwave frequency, 400 MHz $^1H$ frequency setup (Bruker Biospin) at 100 K temperature and 8.5 kHz MAS.

Topological studies: The mobility-edited 2D $^1H(^1H)^{13}C$ experiment[22] was performed at 700 MHz with 16.5 kHz MAS at 300 K temperature using a $T_2$ relaxation filter of 2.5 ms. After dephasing of the rigid signals, magnetization from mobile lipid and water molecules was transferred to the rigid antibiotic with 5 ms $^1H$–$^1H$ mixing, and subsequently transferred to $^{13}C$ nuclei of [R4L10]-teixobactin with a short (200 µs) cross-polarization step.

Structural studies: To probe intermolecular contacts between $^{13}C,^{15}N$-labelled [R4L10]-teixobactin molecules bound to unlabelled ($^{12}C,^{14}N$) Lipid II, we ran 2D $^{13}C^{13}C$ PARIS experiments with magnetization transfer times of 50, 300, 600, and 1000 ms at 950 MHz and 15.5 kHz MAS. To probe interfacial contacts between $^{13}C,^{15}N$-labelled [R4L10]-teixobactin and $^{13}C,^{15}N$-labelled Lipid II, we ran 2D $^{13}C^{13}C$ PARIS experiments with magnetization transfer times of 25, 50, 600, and 1000 ms at 950 MHz and 18 kHz MAS. 1D $^{31}P$ CP experiments were acquired at 500 MHz with 12 kHz MAS. The 2D $^1H^{31}P$ experiment was acquired at 800 MHz magnetic field and 60 kHz MAS using 1.75 ms CP transfer from $^1H$ to $^{31}P$.

Studies of the impact of membrane and receptor composition: 2D NH experiments were conducted with different numbers of scans due to strongly decreased spectral sensitivity with increasing anionic membrane charge.

Relaxation studies: $^1H$-detected $^{15}N$ $T_{1rho}$ relaxation experiments were carried out at 950 and 700 MHz magnetic field, respectively, using 60 kHz MAS[51]. The $^{15}N$ transverse magnetization decay was probed with a $^{15}N$ spinlock field of 17.5 kHz. For the bulk, we measured the intensity of the most intense signal at ~8.80 $^1H$ ppm. $T_{1rho}$ trajectories were fit to single exponentials.

**NMR structure calculation**. Parameterization of [R4L10]-teixobactin: An initial linear [R4L10]-teixobactin topology was generated in CNS 1.2[53]. D-amino acids were generated by inverting relevant dihedral and improper torsion angles. D-Me-Phe N-methyl group parameters were based on monomethyl lysine

(as accommodated by HADDOCK). The rings were defined by manually removing Thr8 hydroxyl proton and Ile11 carboxyl –OH from the topology, and introducing the relevant bond lengths, bond angles, dihedral angles, and improper torsion angles in the topology. Ester bond geometric parameters were based on the crystal structure of a teixobactin analogue (PDB 6E00)[11]; partial charges were based on protonated glutamic acid sidechain (as defined in the OPLS[54] force field used in HADDOCK). A monomeric [R4L10]-teixobactin starting model for HADDOCK structure calculation was then generated in CNS[55], using only dihedral angle restraints for residues Ile2–Ser7 initially obtained from an X-ray structure[11] and validated by our chemical shift analysis (see Supplementary Information).

Parameters for Lipid II were taken from ref. [56].

NMR restraints: NMR structure calculations were performed with HADDOCK version 2.4[23]. SSNMR restraints are described in detail in the Supplementary Information. In total, we used 23 intermolecular [R4L10]-teixobactin–[R4L10]-teixobactin NMR distance restraints to define the fibre-like β-sheet formed by oligomerised teixobactins; 22 intermolecular [R4L10]-teixobactin–Lipid II NMR distance restraints were used to define the antibiotic–receptor interface. The structure of [R4L10]-teixobactin was further defined by 12 dihedral restraints and 7 intramolecular NMR distance restraints. The head-group structure of Lipid II was further defined by 21 NMR distance restraints. See Supplementary Table 3 for a summary of the restraints.

Structure calculation protocol: HADDOCK version 2.4[23] was used for the structure calculations. An eight-body docking (4 Lipid II and 4 [R4L10]-teixobactin molecules) was performed using the distance and dihedral restraints described above. 7000 models were generated in the rigid body docking stage of HADDOCK, of which the best scoring 500 were subjected to the flexible refinement protocol of HADDOCK. The resulting models were finally refined in explicit solvent. Default HADDOCK settings were used except for doubling the weighting of the distance restraints during all stages of the structure calculation. The final models were further filtered based on the topological requirements (i.e., the hydrophobic tails of all four Lipid II molecules must point in the same direction as the membrane-anchoring residues Ile2, Ile5, and Ile6). This resulted in a final ensemble of 26 structures.

Analysis of calculated structures: Structural and violation statistics of the final 26 structures are discussed in detail in the Supporting Information. The average backbone RMSD (from the average structure) of the 26 [R4L10]-teixobactin molecules in the complex was $1.77 \pm 0.49$ Å.

**Sample preparations**. Lipid II was synthesized based on enzymatic lipid reconstitution using the Lipid II precursors UDP-GlcNAc, UDP-MurNAc-pentapeptide, and polyisoprenolphosphate as substrates[16]. Lysine-form UDP-MurNAc-pentapeptide was extracted from *Staphylococcus simulans* 22, whereas the mDAP-form was obtained from *Bacillus subtills* 168. $^{13}C$,$^{15}N$-labelled UDP-GlcNAc and UDP-MurNAc-pentapeptide (lysine form) were extracted from *S. simulans* 22 grown in [$^{13}C$,$^{15}N$]-labelled rich medium (Silantes) and supplemented with [U-$^{13}C$]-D-glucose and [$^{15}N$]-NH$_4$Cl (Supplementary Fig. 14a). Polyisoprenophosphate was synthesized via phosphorylation of polyisoprenol obtained from *Laurus nobilis*[57]. The head-group precursors were extracted from bacteria and polyisoprenol was extracted from leaves as described[58,59]. After synthesis, Lipid II was extracted with 2:1 BuOH:(Pyr/Acetate, 6 M) and purified with a DEAE cellulose resin (acetate-form, BioPHoretics) using a salt gradient (Supplementary Fig. 14b, c). Fractions containing pure Lipid II were pooled, dried, and dissolved in 2:1 chloroform/methanol. Lipid II concentration was estimated through an inorganic phosphate determination[60]. The labelled Lipid II yield was ~1.0 μmol/L. Lipid I was produced in the same manner, but without adding UDP-GlcNAc.

Lipid I or II-doped multi-laminar vesicle preparations: DOPC or DOPG lipids were mixed with Lipid I or II in 1:1 MeOH:CHCl$_3$ at the final Lipid I or II molar ratio (0–4 mol%/mol). The mixtures were dried and the lipid films were hydrated by vortexing with 2 mL of a teixobactin solution in 40 mM phosphate, 25 mM NaCl, pH 7.0.

SSNMR samples: MLVs were collected by centrifugation (20,000×$g$) and loaded into ssNMR rotors. For 3.2 mm rotors, we used 800 nmol of antibiotic with unlabelled Lipid II or I, while we used 400 nmol with labelled Lipid II. For 1.3 mm rotors, samples contained 200 nmol of antibiotic for unlabelled Lipid II or I, while we used 100 nmol for labelled Lipid II. For all ssNMR samples, we used 4 mol %/mol Lipid II or I in the MLVs.

Native membrane vesicle preparations were obtained from *Micrococcus flavus* DSM 1790[13,16]. The amount of antibiotic was ~7.5 and 30 nmol for the 1.3 and 3.2 mm rotors, respectively.

Prior to the DNP measurements, the DNP samples were suspended in 60% glycerol-d8, 35% buffer solution (25 mM NaCl, 15 mM Tris–HCl pH 7.0 final concentration), and 5% 15 mM AMUPol (final concentration) in D$_2$O.

**Synthesis of [R4L10]-teixobactin**. Labelled teixobactins were synthesized using adapted protocols, described in detail in Supplementary Figs. 15, 16, and 17[10]. For simplicity, in this paper D-Arg$_4$-Leu$_{10}$-teixobactin is represented as [R4L10]-teixobactin and Leu$_{10}$-teixobactin as [L10]-teixobactin.

Steps are illustrated in Supplementary Fig. 15: (step a) 2-Chlorotrityl chloride resin (manufacturer's loading = 1.6 mmol/g, 150 mg resin) was swelled in DCM in

a reactor. To this resin was added 4 eq. Fmoc-Ala-OH/8 eq. DIPEA in DCM and the reactor was shaken for 3 h. The loading determined by UV absorption of the piperidine-dibenzofulvene adduct was calculated to be 0.56 mmol/g, (150 mg resin, 0.084 mmol). Any unreacted resin was capped with MeOH:DIPEA:DCM = 1:2:7 by shaking for 1 h. (step b) The Fmoc protecting group was deprotected using 20% piperidine in DMF by shaking for 3 min, followed by draining and shaking again with 20% piperidine in DMF for 10 min. AllocHN-D-Thr-OH was then coupled to the resin by adding 3 eq. of the AA, 3 eq. HATU, and 6 eq. DIPEA in DMF and shaking for 1.5 h at room temperature. (step c) Esterification was performed using 10 eq. of Fmoc-Ile-OH, 10 eq. DIC, and 5 mol% DMAP in DCM and shaking the reaction for 2 h. This was followed by capping the unreacted alcohol using 10% Ac$_2$O/DIPEA in DMF shaking for 30 min and Fmoc was removed using protocol described earlier in step (b). (step d) Fmoc-Leu-OH was coupled using 4 eq. of AA, 4 eq. HATU, and 8 eq. DIPEA in DMF and shaking for 1 h followed by Fmoc deprotection using 20% piperidine in DMF as described earlier. (step e) The N terminus of Leu was protected using 10 eq. Trt-Cl and 15% Et$_3$N in DCM and shaking for 1 h. Protection was verified by Ninhydrin colour test. (step f) The Alloc protecting group of D-Thr was removed using 0.2 eq. [Pd (PPh$_3$)]$^0$ and 24 eq. PhSiH$_3$ in dry DCM under argon for 20 min. This procedure was repeated and the resin was washed thoroughly with DCM and DMF to remove any excess Pd stuck to the resin (step g). All amino acids were coupled using 4 eq. amino acid, 4 eq. HATU, and 8 eq. DIPEA for 1 h. (step h) The peptide was cleaved from the resin without cleaving off the protecting groups of the amino acid sidechains using TFA:TIS:DCM = 2:5:93 and shaking for 1 h. (step i) The solvent was evaporated and the peptide was dissolved in DMF to which 1 eq. HATU and 10 eq. DIPEA were added and the reaction was stirred for 30 min to perform the cyclisation. (step j) The sidechain protecting groups were then cleaved off using TFA:TIS:H$_2$O = 95:2.5:2.5 by stirring for 1 h. The peptide was precipitated using cold Et$_2$O (−20 °C) and centrifuging at 7000 rpm to obtain a white solid. This solid was further purified using RP-HPLC using the protocols described in Supplementary Fig. 16c, d[10]. The teixobactin analogues (1–2) were identified by MS in positive mode (Supplementary Table 4 and Supplementary Fig. 16e, f).

**Reporting summary**. Further information on research design is available in the Nature Research Reporting Summary linked to this article.

## Data availability

Data supporting the findings of this manuscript are available from the corresponding author upon reasonable request. A reporting summary for this Article is available as a Supplementary Information file. The solid-state NMR assignments of [R4L10]-teixobactin and Lysine-Lipid II have been deposited in the BMRB (accession number 50202). The coordinates of the [R4L10]-teixobactin–Lipid II complex have been deposited in the PDB database PDB 6YFY. The source data underlying Figs. 1d, 2e, 4c, and Supplementary Figs. 2b, 3a–c, 10a, 11b, c are provided as a Source Data file.

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

## Acknowledgements

This work is part of the research programmes ECHO, TOP, TOP-PUNT, VICI, and VIDI with project numbers 723.014.003, 711.018.001, 700.26.121, 700.10.443, and 718.015.00, which are financed by the Dutch Research Council (NWO). I.S. acknowledges the Innovate UK and Department of Health and Social Care (DHSC), UK and Rosetrees Trust for their kind support (SBRI grant 106368-623146 and Rosetrees Trust grant JS15/M783). The views expressed in this publication are those of the authors and not necessarily those of Innovate UK or DHSC, UK. Experiments at the 950 MHz instrument were supported by uNMR-NL, an NWO-funded Roadmap NMR Facility (no. 184.032.207). NMR measurements at the 800 MHz instrument at the University of Florence were supported by iNEXT (project number 653706), a Horizon 2020 programme of the European Union. The HADDOCK software development is supported by the European Union Horizon 2020 BioExcel project numbers 675728 and 823830.

## Author contributions

R.S., J.M.-S., A.L.P., B.J.A.V., M.B., M.L., and M.W. did and analysed NMR experiments; J.M.-S., R.S., and E.B. prepared ssNMR samples; I.S. conceived the design and synthesis of teixobactins; A.P., S.D., and I.S. synthesized teixobactins; R.S., J.L., and E.B. did fluorescence measurements; J.M.-S. and E.B. prepared Lipid II; B.J.A.V., A.M.J.J.B., and M.W. did structure calculations; R.S., S.J., E.J.A.V., E.B., and M.W. did and analysed ITC experiments. All authors wrote and edited the manuscript.

## Competing interests

The authors declare no competing interests.
