## [Peer Review File · Nature Communications]

Peer Review File

Reviewers' comments first round:

Reviewer #1 (Remarks to the Author):

This is an impressive paper that should be published after revision. The authors perform detailed NMR-based structural studies of a teixobactin analogue (Arg4,Leu10-teixobactin) and characterize its interactions with lipid II. Distance restraints from the NMR studies allow a molecular model for the interaction with the lipid sugars and pyrophosphate to be created, as well as molecular model for the necessary self-assembly of the teixobactin analogue. These results are interpreted in light of a known crystal structure and SAR studies.

Additional studies with lipids derived from a model organism (*M. flavus*) in anionic membranes show weaker binding than expected. Based on this observation and additional NMR studies suggest a model in which teixobactin forms aggregates on the cell surface. The authors also observe its aggregation on vesicles through fluorescence microscopy.

The title of the paper is misleading, for two reasons, claiming teixobactins and cellular conditions. These statements should be replaced with more specific language, such as "a teixobactin analogue" or "teixobactin analogues" and "with lipid membranes." The analogues the authors studied, Arg4,Leu10 and Leu10-teixobactin, lack the native enduracididine at position 10. Using "teixobactin analogues" will clarify to the reader that the authors used analogues that lack enduracididine.

Figure 1g: a negative control is recommended for the fluorescence microscopy experiment to confirm if the lipid II clusters reflect sequestration due to antibiotic action. An inactive teixobactin analogue, such as an N-terminally truncated teixobactin or an acyclic teixobactin, should not result in micron-sized clusters.

Figure 3c and Lines 203, 214-215: change "D-N-Met-F1" to "D-N-Me-F1" since Met can be confused with methionine and since Me is a common abbreviation for methyl.

Figure 4a: if possible, cell membranes from another Gram-positive bacterium should be used to further support 2D 15N1H ssNMR data.

Line 302: specify that *M. flavus* is a Gram-positive bacteria

Line 311: should be rationale, not rational

Line 506: ideally these measurements should be done in triplicate.

Line 542: specify the emission range for fluorescence detection.

Morris et al. very recently reported the molecular localization of a fluorescently labeled teixobactin analogue in Gram-positive bacteria using fluorescence microscopy (ACS Chem. Biol. 2020, Just Accepted). The imaging studies therein show that the teixobactin analogue has two staining patterns: (1) teixobactin aggregates that result in whole-cell staining in the absence of polysorbate 80; and (2) lateral and septal staining of cells in the presence of polysorbate 80. The whole-cell staining observed in the absence of polysorbate 80 is congruent with the authors' findings that the teixobactin analogue forms micron-sized clusters. This paper could be cited in support of the idea of clustering of PG precursor molecules in the membranes of bacteria.

Reviewer #2 (Remarks to the Author):

Dr. Weingarth and coworkers have provided a high-resolution view of a complex in which teixobactins is binding either model membranes or cellular membranes. The authors have found

that teixobactins capture lipid II in micrometer-sized clusters on membrane surfaces and proposed a structural model of teixobactin-lipid II complex. Another major and surprising finding is the weak/moderate interactions between teixobactin and lipid II interactions in cellular membranes, which suggests that direct binding to cell wall precursor is not the only mechanism for antimicrobial activity.

Information such as that was unavailable and has the great potential of revising our view of the functional mechanism of teixobactins and their analogues. The NMR data is of high quality and supports the conclusions. Below are multiple minor concerns that could help improve the manuscript further.

1) Some strong statements need re-consideration.

First, in the abstract, the authors state that "direct interaction with cell wall precursors is not the principal killing mechanism." With the current data, it will be more accurate to claim that it is "not the sole mechanism."

Second, a minor concern is the statement of "cellular condition" in the title, figure captions, and the text. As what is used in the study is actually membrane vesicles prepared from *M. Flavus*, it might be more specific and more appropriate to use "cellular membranes" instead.

2) The manuscript needs a dedicated Discussion section to summarize our current knowledge as well as the ambiguous aspects of the action mode of teixobactins, which should be coupled with a discussion of the novelty of the findings in this study. This part of information is currently missing in the writing.

3) The experimental data, especially the solid-state NMR data, are of high quality and are presented well in the maintext and the Extended dataset. The extended data will need to be modified to Supplementary Information and follow the format and size requirement.

Also, there are a few technical details to be clarified:

Line 573: it will be helpful to put down the rationale for choosing a slow spinning for the TOBSY experiment.

Line 625: what is the microwave power of the gyrotron used for the DNP experiment?

4) Minor format issues such as inconsistency in the use of units or journal name and page number: Line 353, 420, 427, 429, 434, 530, 553, 585, 909, 914, 925 (please check through the manuscript)

Also, Line 123, Extended data Fig. 4 showed up before Fig. 2-3. Please Adjust the order.

Response to the Reviewers' Comments on the Manuscript NCOMMS-20-06224-T entitled "Mode of action of teixobactins in cellular conditions" submitted to Nature Communications

We would like to thank the Reviewers for carefully reading our paper and for their critique. We are very encouraged by the very positive evaluation of our manuscript by both Reviewers. Their valuable comments have allowed us to improve our manuscript. In the following, we include a point-by-point response to the questions and comments of each Reviewer.

Reviewer #1:

This is an impressive paper that should be published after revision. The authors perform detailed NMR-based structural studies of a teixobactin analogue (Arg4,Leu10-teixobactin) and characterize its interactions with lipid II. Distance restraints from the NMR studies allow a molecular model for the interaction with the lipid sugars and pyrophosphate to be created, as well as molecular model for the necessary self-assembly of the teixobactin analogue. These results are interpreted in light of a known crystal structure and SAR studies.

Additional studies with lipids derived from a model organism (*M. flavus*) in anionic membranes show weaker binding than expected. Based on this observation and additional NMR studies suggest a model in which teixobactin forms aggregates on the cell surface. The authors also observe its aggregation on vesicles through fluorescence microscopy.

We are very grateful for her/his very positive opinion on our manuscript!

1. The title of the paper is misleading, for two reasons, claiming teixobactins and cellular conditions. These statements should be replaced with more specific language, such as "a teixobactin analogue" or "teixobactin analogues" and "with lipid membranes. The analogues the authors studied, Arg4,Leu10 and Leu10-teixobactin, lack the native enduracididine at position 10. Using "teixobactin analogues" will clarify to the reader that the authors used analogues that lack enduracididine.

We thank the reviewer very much for her/his valuable criticism. We concur with Reviewer 1 that 'cellular conditions' could be misunderstood. However, "lipid membranes" would be misleading, given that we used cellular membranes that contain all its naturally occurring components (lipids, proteins, and other biomolecules). This

preparation highly contrasts with pure lipid extracts. Therefore, we suggest changing the title to

"Mode of action of teixobactins in cellular ~~conditions~~ membranes"

This new title is also suggested by Reviewer 2, and is consistent with the title of our previous, well-received Nature Communications article (*'High-resolution NMR studies of antibiotics in cellular membranes'*).

Regarding the use of teixobactins in the title:

We thank Reviewer 1 very much for her/his comment. We would humbly request to keep teixobactins in the title for the following arguments:

- Teixobactin is generally referred to as a new class of antibiotics. There are numerous examples (e.g., Refs. ¹⁻⁴) in the scientific literature, including recent Nature ChemBio/Nature Communications articles, that refer to teixobactin as a class of antibiotics
- The term teixobactins has been used in a number of publications, including Nature journals (e.g., Refs. ¹⁻³)
- Therefore, in light of the accepted use of teixobactins in previous publications, we do believe that a shorter title will enhance the accessibility of our manuscript to the broad readership of Nature Communications

2. Figure 1g: a negative control is recommended for the fluorescence microscopy experiment to confirm if the lipid II clusters reflect sequestration due to antibiotic action. An inactive teixobactin analogue, such as an N-terminally truncated teixobactin or an acyclic teixobactin, should not result in micron-sized clusters.

We thank the Reviewer very much for her/his recommendation. We deeply regret that we were not able to conduct additional microscopy studies before the closure of our University due to SARS-CoV-2. We would hope that the microscopy data that we show in the manuscript, for two different teixobactins, provide solid evidence for cluster formation. Furthermore, as emphasized by Reviewer 1 (see point 9. below), the cluster formation that we describe is corroborated by recent microscopy studies by Morris et al. (ACS Chem. Biol. 2020).

3. Figure 3c and Lines 203, 214-215: change "D-N-Met-F1" to "D-N-Me-F1" since Met can be confused with methionine and since Me is a common abbreviation for methyl.

We followed the Reviewer's valuable suggestion and changed '*D-N-Met-F1*' to '*D-N-Me-F1*' throughout the manuscript, including Figure 3c.

4. Figure 4a: if possible, cell membranes from another Gram-positive bacterium should be used to further support 2D $^{15}\text{N}/^1\text{H}$ ssNMR data.

[Redacted]

[Redacted]

5. Line 302: specify that *M. flavus* is a Gram-positive bacteria

We thank the Reviewer for her/his efforts to enhance the accessibility of our manuscript. We added on page 12, l. 314 of the revised version:

*'Here, we investigated the [R4L10]-teixobactin - Lipid II complex directly in natural membranes isolated from the Gram-positive bacterium *M. flavus*.'*

6. Line 311: should be rationale, not rational

We thank the reviewer for her/his attentiveness. We corrected it.

7. Line 506: ideally these measurements should be done in triplicate.

We are grateful for the Reviewer's comment. Meanwhile, we have run all reported ITC measurements in triplicate and measured K_d were all well reproducible (see Figure R2 below). We added on page 17, l. 452 of the revised version:

"All experiments were performed in ~~duplicates~~ triplicate for each system."

Figure R2: ITC data of [R4L10]-teixobactin and Lys-Lipid II in A-C) neutral DOPC liposomes and D-F) anionic membranes, composed of a 1:3 (mol/mol) DOPG:DOPC mixture. Runs A,B,D,E) were done with Low Volume NanoITC (TA Instruments-Waters LLC), while runs C,F) were done with the PEAQ-ITC (Malvern). G,H) show the background (drug added to Lipid II-free liposomes). Due to a difference in data representation of exothermic plots between the TA and Malvern machines, graphs C,F) show downward peaks for the same reaction.

Binding to Lipid II in DOPC liposomes: average $K_d = 1.72 \times 10^{-7} M \pm 3.61 \times 10^{-9}$
 Binding to Lipid II in DOPC/PG liposomes: average $K_d = 1.79 \times 10^{-5} M \pm 9.02 \times 10^{-6}$

8. Line 542: specify the emission range for fluorescence detection.

We are grateful for the Reviewer's suggestion. We added on page 18, l. 479 of the revised version:

'The NBD label appeared green upon excitation by the 488 nm laser. The emission range for detection was 530nm to 545nm (λ_{em} peak = 539nm).'

9. Morris et al. very recently reported the molecular localization of a fluorescently labeled teixobactin analogue in Gram-positive bacteria using fluorescence microscopy (ACS Chem. Biol. 2020, Just Accepted). The imaging studies therein show that the teixobactin analogue has two staining patterns: (1) teixobactin aggregates that result in whole-cell staining in the absence of polysorbate 80; and (2) lateral and septal staining of cells in the presence of polysorbate 80. The whole-cell staining observed in the absence of polysorbate 80 is congruent with the authors' findings that the teixobactin analogue forms micron-sized clusters. This paper could be cited in support of the idea of clustering of PG precursor molecules in the membranes of bacteria.

We thank the Reviewer very much for her/his constructive advice. We have read Morris' recent paper, which came out after our submission, with great curiosity. We were very excited to see that the observed pronounced staining of cell wall synthesis sites aligns well with our findings on micron-sized cluster formation. We cite Morris' elegant paper as reference 44 and discuss it on p. 15 l. 383:

'This is an additional mode of action, and potentially even the more relevant one in physiological conditions. The relevance of cluster formation for activity is also congruent with studies of fluorescent teixobactins that showed pronounced staining of cell wall synthesis sites in Gram-positive bacteria.'^{44,}

Reviewer #2:

Dr. Weingarth and coworkers have provided a high-resolution view of a complex in which teixobactins is binding either model membranes or cellular membranes. The authors have found that teixobactins capture lipid II in micrometer-sized clusters on membrane surfaces and proposed a structural model of teixobactin-lipid II complex. Another major and surprising finding is the weak/moderate interactions between teixobactin and lipid II interactions in cellular membranes, which suggests that direct binding to cell wall precursor is not the only mechanism for antimicrobial activity.

Information such as that was unavailable and has the great potential of revising our view of the functional mechanism of teixobactins and their analogues. The NMR data is of high quality and supports the conclusions. Below are multiple minor concerns that could help improve the manuscript further.

We are very grateful for her/his very positive opinion on our manuscript!

Minor points:

1) Some strong statements need re-consideration:

A. First, in the abstract, the authors state that “direct interaction with cell wall precursors is not the principal killing mechanism.” With the current data, it will be more accurate to claim that it is “not the sole mechanism.”

We fully agree with the Reviewers criticism. We modified the abstract as she/he suggested:

“Unexpectedly, we found that teixobactins only weakly bind to Lipid II in cellular membranes conditions, implying that the direct interaction with cell wall precursors is not the principal sole killing mechanism. Instead, Our data suggest that the excellent activity of teixobactins is also afforded by an alternative killing mechanism, in which teixobactins block the cell wall biosynthesis by capturing precursors in μm -sized clusters on membrane surfaces.”

B. Second, a minor concern is the statement of “cellular condition” in the title, figure captions, and the text. As what is used in the study is actually membrane vesicles prepared from *M. Flavus*, it might be more specific and more appropriate to use “cellular membranes” instead.

We concur with the Reviewers criticism and heeded her/his valuable suggestion. We changed the title to

‘Mode of action of teixobactins in cellular conditions membranes’

2) The manuscript needs a dedicated Discussion section to summarize our current knowledge as well as the ambiguous aspects of the action mode of teixobactins, which should be coupled with a discussion of the novelty of the findings in this study. This part of information is currently missing in the writing.

We thank the Reviewer for her/his instructive guidance on the format! We have adapted the manuscript to the Nature Communications format. Therefore, we extended the introduction on page 3, and we added a dedicated discussion on page 15.

3) The experimental data, especially the solid-state NMR data, are of high quality and are presented well in the main text and the Extended dataset. The extended data will need to be modified to Supplementary Information and follow the format and size requirement.

We thank the Reviewer for her/his advice. We consistently changed the wording 'Extended Data Fig.' to 'Supplementary Figure' throughout the manuscript and Supplementary Information. We also made sure that the Supporting Information follows the format and size requirements.

3b) Also, there are a few technical details to be clarified: Line 573: it will be helpful to put down the rationale for choosing a slow spinning for the TOBSY experiment.

We thank the Reviewer for her/his suggestion, which will enhance the clarity of our experimental setup. We included an explanation for the choice of a slow spinning frequency for the TOBSY experiment in the Supporting Information in the caption of Supporting Figure 6:

'In our TOBSY sequence, the basic recoupling pulse cycle ($90_x - 360_x - 270_x$) needs to be repeated three times during one rotor period in order to suppress anisotropic NMR interactions. This led to a ^{13}C power requirement of six times the MAS frequency. Therefore, we used slow MAS (8 kHz) for this experiment.'

3c) Line 625: what is the microwave power of the gyrotron used for the DNP experiment?

We are grateful for the Reviewer's comment. We added on page 18, l. 510 of the revised version:

'DNP-enhanced 2D $^{13}\text{C}^{13}\text{C}$ spin diffusion (20 ms mixing time) ssNMR experiments⁵ were carried out using a 263 GHz microwave frequency, 400 MHz ^1H frequency setup (Bruker Biospin) at 100 K temperature and 8.5 kHz MAS'

4) Minor format issues such as inconsistency in the use of units or journal name and page number: Line 353, 420, 427, 429, 434, 530, 553, 585, 909, 914, 925 (please check through the manuscript). Also, Line 123, Extended data Fig. 4 showed up before Fig. 2-3. Please adjust the order.

We thank the Reviewer for her/his attentiveness. We have i) corrected the inconsistencies and ii) adjusted the order of the Supplementary Figures.

References

- 1 Johnston, C. W. *et al.* Assembly and clustering of natural antibiotics guides target identification. *Nat Chem Biol* **12**, 233-239, doi:10.1038/nchembio.2018 (2016).
- 2 Bozhuyuk, K. A., Micklefield, J. & Wilkinson, B. Engineering enzymatic assembly lines to produce new antibiotics. *Curr Opin Microbiol* **51**, 88-96, doi:10.1016/j.mib.2019.10.007 (2019).
- 3 Iyer, A., Madder, A. & Singh, I. Teixobactins: a new class of 21st century antibiotics to combat multidrug-resistant bacterial pathogens. *Future Microbiol* **14**, 457-460, doi:10.2217/fmb-2019-0056 (2019).
- 4 Zong, Y. *et al.* Gram-scale total synthesis of teixobactin promoting binding mode study and discovery of more potent antibiotics. *Nat Commun* **10**, 3268, doi:10.1038/s41467-019-11211-y (2019).

REVIEWERS' COMMENTS second round:

Reviewer #1 (Remarks to the Author):

I am satisfied with the authors' thoughtful efforts to address our concerns and those of the other reviewer, particularly in light of the COVID-19 shutdown, and recommend publication as is. The PDB structure will be a particularly valuable contribution and is a good addition.

James S. Nowick

Reviewer #2 (Remarks to the Author):

The authors has addressed all my concerns. In particular, I would like to mention that the use of "cellular membranes" in the current title is appropriate. It also connects to the previous studies by the authors using cellular membrane systems. The ambiguous aspects of the technical details are now clarified. The maintext and Supplementary Information files have also been reshaped.

The revised manuscript is clear, novel, and technically sound.